# DNase-1 Treatment Exerts Protective Effects in Neurogenic Pulmonary Edema via Regulating the Neutrophil Extracellular Traps after Subarachnoid Hemorrhage in Mice

**DOI:** 10.3390/jcm11154349

**Published:** 2022-07-27

**Authors:** Xinyan Wu, Yinghan Guo, Hanhai Zeng, Gao Chen

**Affiliations:** Department of Neurosurgery, The Second Affiliated Hospital of Zhejiang University School of Medicine, Hangzhou 310009, China; 3150103621@zju.edu.cn (X.W.); guoyinghan36@163.com (Y.G.); 11918362@zju.edu.cn (H.Z.)

**Keywords:** DNase-1, subarachnoid hemorrhage, neurogenic pulmonary edema, neutrophil extracellular traps, inflammation

## Abstract

It has been reported that neutrophil extracellular traps (NETs) involve inflammation, coagulation and cell death. Acute lung injury is also considered to be connected with NETs. Deoxyribonuclease I (DNase-1), a clinical medication for the respiratory system, has been reported to degrade cell-free DNA (cfDNA), which is the main component of NETs. Herein, we did research to clarify the therapeutic value of DNase-1 in NPE after SAH. In this model, we found that the treatment of DNase-1 remarkably decreased lung water, neutrophilic infiltration and inflammation. In addition, DNase-1 inhibited the NETs and proinflammatory subtype transition of the macrophages. Moreover, the depletion of neutrophil also verified the role of NETs in NPE. Our results suggest that DNase-1 has the potential to effectively relieve the NPE after SAH and to be a clinical drug for use after SAH.

## 1. Introduction

Subarachnoid hemorrhage (SAH) accounts for 5% of strokes and often occurs at a relatively young age [1]. The final stage for the patients with SAH remains poor, with the mortality rates reaching 45% and significant morbidity among survivors [2]. A lot of systemic complications follow a subarachnoid hemorrhage, which are primarily caused by the activation of the sympathetic nervous system and have a negative effect on survival [3]. Neurogenic pulmonary edema (NPE), one of the most important systemic complications, is considered to be related to the decline in survival [4,5,6].

NPE is usually defined as an acute pulmonary edema occurring shortly after a central neurologic insult [4]. Although there is a wide range of research on NPE, the mechanism of it is still poorly understood [7]. There have been two theories that may explain the association of pulmonary edema with a CNS injury [8]. One theory is that the injury of the central regulatory center may affect the pulmonary vascular bed by altering the pulmonary autonomic nervous system (ANS). The stimulation of vasomotor centers in the central nervous system in excess may be another representative explanation. Otherwise, the recruitment of neutrophils associated with inflammatory reactions can maintain and even aggravate the edema by enhancing the permeability of capillary [9].

Neutrophil extracellular traps (NETs), one of the pathogenic mechanisms for neutrophils, are structures of chromatin filaments coated with histones, proteases and granular and cytosolic proteins [10]. NETs seem to have a protective effect by trapping the pathogenic microorganisms to prevent proliferation. However, NETs have also been proven to have a negative effect by triggering coagulation, inflammation and cell death [11]. The specific relationship between NETs and noninfectious respiratory disease is still not totally clear. In vitro research has shown that the formation of NETs induces the death of lung epithelial and endothelial cell lines [12], which means NETs play a directly toxic role in the alveolar–capillary barrier [13].

Deoxyribonuclease I (DNase-1), a clinical medication for the respiratory system [14], can degrade cell-free DNA (cfDNA), which is the main component of NETs [15]. There is a growing number of evidence that confirms the decrease of NETs can reduce body damage [16]. Moreover, there has been research suggesting that DNase-1 therapy reduces the inflammatory response, including NETs, without increasing the risk of bleeding [17]. In this study, we attempted to evaluate the therapeutic value of DNase-1 for NPE in a mouse model of SAH.

## 2. Materials and Methods

### 2.1. Animals

Adult male C57BL/6 mice (22–25 g) were acquired from Shanghai SLAC Laboratory Animal Co., Ltd. The animals were fed on a 12-h light/dark cycle under stable temperature and humidity conditions. All procedures were in strict accordance with the guidelines of the ‘Principles of Laboratory Animal Care’ (NIH publication No. 80-23, revised 1996) and were approved by the Institutional Animal Care and Use Committee of Zhejiang University.

### 2.2. Study Design

Two experiments were involved in this study. First, the mice were randomly divided into a sham group, SAH + vehicle group, and SAH + DNase-1 group. The sham group obtained the vehicle and was subjected to a process analogous to that of the SAH + vehicle group but without perforation. Second, the mice were randomly assigned to the SAH + vehicle group and SAH + anti-Ly6G group. According to the previous studies, we deemed to investigate the endpoints of this research at 24 h after SAH [18].

### 2.3. Drug Administration

DNase-1 (50 μg in 250 μL of normal saline intraperitoneally and 10 μg in 250 μL of normal saline as a second dose intravenously) or the vehicle was injected into mice bodies 3 h after SAH induction, as the previous research described [19]. The sham group and the SAH + vehicle group were injected with the same volume of vehicle in the corresponding way at the corresponding time points.

Fifty micrograms of anti-Ly6G antibodies (Thermo Fisher, Waltham, MA, USA) were injected into mice at 2 days before SAH intravenously in order to eliminate the neutrophils [20]. The SAH + vehicle group was injected with the same volume of the vehicle at the corresponding time points.

### 2.4. Mice SAH Model

The SAH model was performed using the endovascular perforation technique [21]. Briefly, after the mice were anesthetized, the left carotid artery and its branches were exposed. A 5-0 monofilament nylon suture was placed in the external carotid artery and advanced through the internal carotid artery until resistance, approximately 10–11 mm from the common carotid artery bifurcation. The suture was further advanced to perforate the internal carotid artery near the bifurcation. The severity of the SAH was quantified using a previously reported grading scale [22]. A total score ranging from 0 to 18 was measured.

### 2.5. Mortality and Neurological Deficits

Mortality and neurological scores were calculated 24 h after SAH. The scoring system of Garcia et al. was used to evaluate the neurological scores [23]. Briefly, the evaluation consists of six tests that can be scored 0–3 or 1–3, and the final scores ranged from 3 to 18. A lower score represents serious neurological deficits. All of the tests were evaluated by a researcher who was blind to the treatment.

### 2.6. The NPE Criteria

The NPE was diagnosed by wheezing, a pink froth discharge from the snout and sporadic pulmonary hemorrhagic lesions after biopsy [24].

The wet-to-dry weight ratio was measured on the left lung lobes, as mentioned previously [25]. Generally, we firstly dissected the tissue samples free from the nonparenchymal tissue. The dissected lung samples of each animal were placed in a dish and weighed. Afterwards, the samples were dried for 24 h at 65 °C and weighed again. The whole drying–weighing procedure was repeated until no weight change could be detected over a 24-h period. At this time point, the samples were identified to be dry. The lung water was expressed as a wet-to-dry weight ratio.

### 2.7. Lung Histopathology and Immunofluorescent Staining

The mice were sacrificed and perfused intracardially with PBS and 4% paraformaldehyde. Then, the lung tissue was soaked in 4% formaldehyde for 24 h at 4 °C and then steeped in 30% sucrose solution until the tissue sank (about 48 h). The lungs were cut into 8-um-thick slices. The lung slides were stained with hematoxylin and eosin (H&E) and examined using light microscopy [25]. For immunofluorescent staining, the slides were preincubated in blocking solution. Then, the lung sections were incubated for over 12 h at 4 °C with the primary antibodies: anti-CitH3 (ab5103, Abcam, Cambridge, UK), anti-Ly6g (ab25377, Abcam), anti-NE (ab68672, Abcam), CD16 (PA5-47230, Invitrogen, Waltham, MA, USA) and CD86 (BS-1035R, Bioss, Woburn, MA, USA). Then, the lung sections were incubated with secondary antibodies at room temperature for 2 h. At last, the sections were covered with DAPI and observed using a fluorescence microscope. At least three sections per mice were examined.

### 2.8. qRT-PCR

The lung specimen was collected for detection at 24 h after SAH. The total mRNA was then extracted using the TRIzol^TM^ Plus RNA Purification Kit. Subsequently, the total RNA from each sample was used to synthesize cDNA using Primescript RT Master Kit (Takara, RR420A, Kusatsu, Japan) according to the manufacturer’s instructions. Then, the SYBR Premix Ex Taq™ Kit (Takara, RR036A) was used for real-time PCR. The relative mRNA level of each target gene was calculated using the 2^−∇∇CT^ methods, as previously described [26]. The primers used for qRT-PCR are listed in Appendix A.

### 2.9. Flow Cytometric Analysis

The blood was analyzed by a flow cytometric analysis (FACS) after erythrolysis. The antibodies used were as follows: CD45-PerCp/Cy5.5(103132, 1:400, Biolegend, San Diego, CA, USA), CD11b-APC (101212, 1:400, Biolegend) and Ly6G-PE (127608, 1:400, Biolegend). The percentage of neutrophil represents the effect of the Ly6G antibody in mice.

### 2.10. Statistical Analysis

Continuous data were shown as the mean ± standard deviation (SD) or median (interquartile range) based on the normality and homogeneity of variance. For data conforming to normality, we analyzed the significant differences among groups by using the Student’s *t*-test or one-way analysis of variance (ANOVA). A nonparametric test or Kruskal–Wallis test was used to analyze the significant differences among groups where the data failed to normality. A *p*-value < 0.05 indicated statistical significance. GraphPad Prism (Graphpad Software, San Diego, CA, USA) and SPSS software (Version 23.0, IBM, Armonk, NY, USA) were used for the statistical analysis.

## 3. Results

### 3.1. DNase-1 Treatment Relieves the Injury of NPE after SAH

For all SAH mice, the incidence rate of NPE was 62.18% (*n* = 74/119). In the first experiment, the sham-operated mice were all alive. For SAH mice with NPE, the mortality of the SAH + vehicle group was 29.03% (*n* = 9/31), while the mortality of the SAH + DNase-1 group was 24.14% (*n* = 7/29). In the second experiment, the mortality was 25.00% (*n* = 2/7) in the SAH + vehicle group, while the mortality was 25.00% (*n* = 2/7) in the SAH + anti-Ly6G group. There was no subarachnoid blood clot in the sham-operated mice, and compared with the SAH + vehicle group, the SAH + DNase-1 group had no significant increase in SAH score (*p* > 0.05, Figure 1A). The modified Garcia score of SAH + vehicle mice was significantly lower than that of the sham-operated mice. SAH + DNase-1 mice showed greater improvements in neurological deficits at 24 h after SAH compared with the SAH + vehicle mice (*p* < 0.05, Figure 1B).

The wet weight/dry weight (W/D) ratio, a parameter of the lung water contents, increased significantly in the SAH + vehicle group in comparison to the sham-operated group (*p* < 0.05, Figure 1C). Compared with the SAH + vehicle group, the DNase-1 treatment group showed a significant reduction in the W/D ratio (*p* < 0.05, Figure 1C).

The H&E-stained sections of the SAH + vehicle group mice showed interstitial edema and neutrophil infiltration in the lung tissue, whereas the sham group exhibited almost normal structure. DNase-1 treatment ameliorated the significant increase of alveolar interstitial edema and neutrophil infiltration caused by SAH (Figure 1D).

The data from qRT-PCR showed a significant increase in the level of TNF-α, IL-1β, IL-6 and IL-10 after SAH (*p* < 0.05, Figure 1E–H). Under DNase-1 administration, the upregulated level of TNF-α, IL-1β and IL-6 was then remarkably inhibited, whereas the expression level of IL-10 was further upregulated (*p* < 0.05, Figure 1E–H).

### 3.2. DNase-1 Treatment Inhibits NETs in NPE after SAH

We used several markers to represent NET formation, such as CitH3 and NE. There were almost no CitH3-Ly6G-positive cells and NE-Ly6G-positive cells detected in the lung tissue of the sham-operated group (Figure 2A,B). In comparison to the sham-operated group, the NETs in the lung tissue in the SAH + vehicle group increased significantly, whereas the DNase-1 treatment after SAH significantly inhibited the level of NETs in the lung tissue (*p* < 0.05, Figure 2A–D and Appendix A).

### 3.3. DNase-1 Treatment Reduces the Proinflammatory Subtype Transition of Macrophage in NPE after SAH

We use two markers (CD16 and CD86) to represent the proinflammatory subtype transition of the macrophage. The sham-operated group showed a few CD16-positive cells and CD86-positive cells in the lung tissue (Figure 3A,B). Compared with the sham-operated group, the CD16-positive cells and CD86-positive cells in the SAH + vehicle group increased significantly, while the DNase-1 treatment after SAH significantly reduced the level of CD16-positive cells and CD86-positive cells in the lung tissue (*p* < 0.05, Figure 3A,B).

### 3.4. NETs Are Possible Pathogenic Mechanism of NPE after SAH

To further determine the role of neutrophils in NPE after SAH, Ly6G antibodies were used. FACS presented the effective role of the Ly6G antibody in the depletion of neutrophils (Figure 4A). Compared with SAH + vehicle mice, the injection of the Ly6G antibody showed a significant inhibition in the increase of NETs in the lung tissue after SAH (*p* < 0.05, Figure 4B–D). These results support the view that the release of decondensed DNA of neutrophils and the formation of NETs are the pathogenic process of NPE after SAH.

## 4. Discussion

Increased pulmonary capillary permeability due to SAH is considered to be related to neutrophil recruitment, suggesting the role of NETs [10,27]. In this article, we reported the existence of NETs in the mouse lungs during the SAH and showed that the treatment based on DNase-1 can reduce the formation of NETs, decrease the proinflammatory response, cut back the neurological deficits and improve the alveolar interstitial edema. Consequently, our results provided the evidence that DNase-1 or interventions targeting NETs can ease NPE due to SAH. It also provided support for possible clinical applications in the future.

There are at least three rodent models for studying SAH, including injecting autologous blood directly into the cisterna magna, perforating an intracranial bifurcation of the intracranial artery, etc. [28]. In our research, we chose the endovascular perforation model to study the NPE after SAH. On one hand, 85% of the clinical SAH results from the spontaneous rupture of a cerebral aneurysm and, thus, the endovascular perforation model is the closest imitation of human SAH [28,29]. This model is more conducive to the implementation of a clinical translational application. On the other hand, the previous studies focusing on the NPE after SAH preferred to create this model for research purposes [24,25,30]. In addition to these reasons, the mortality of SAH mice (29.03% versus 44%) and the incidence of NPE after SAH (62.18 versus 69.57%) in our research were nearly consistent with the related references [24,31]. In summary, the model we chose for our research purpose was feasible and reliable.

The current therapy targeted on NPE focuses on the treatment of neurologic conditions and supportive treatment for pulmonary edema [32]. The supportive treatment includes volume management and ventilation strategies, but both of them have limitations. Volume management relies on vasoactive drugs, diuretics, fluid supplements and so on [4,33,34]. However, although adequate volumes are essential for brain resuscitation, the management of NPE may require effective volume reduction, which means that the balance of the volume control remains a challenge [32]. Ventilation strategies also face a dilemma because of the different goals of carbon dioxide control in a brain injury and acute lung injury [32]. Thus, the specific treatment strategy criteria of NPE is still uncertain.

More and more studies have shown that NETs are related to acute injury and coagulation, not just limited to its role in infection [10]. Moreover, it has been reported that the accumulation of NETs in the lungs leads to lung injury [35]. NETs have a proinflammatory effect on airway epithelial cells and, even further, promote the recruitment of neutrophils, leading to persistent inflammatory lung injury [36,37]. In our study, we found that lung water assessed as W/D was significantly increased after 24 h of SAH in mice. Meanwhile, we detected neutrophil infiltration and NET formation in the lung tissue. These findings prove that NETs are involved in the NPE caused by SAH.

Previous research has shown that DNase-1 can regulate NET formation and thereby degrade it. DNase-1 targeting the components of NETs plays a part in transfusion-related acute lung injury and is also effective for the patients with acute lung injury [38]. We firstly evaluated the effect of DNase-1 on NPE after SAH. Based on the previous report, we chose to inject DNase-1 (50 μg in 250 μL of saline intraperitoneally and a second dose of 10 μg intravenously) 3 h after SAH induction. Encouragingly, the assessed parameters of systemic inflammation in the lungs have improved the downregulated levels of TNF-α, IL-1β and IL-6 and the upregulation of IL-10. To explore the possible mechanism, we found that proinflammatory subtype transition may attend the process. The involvement of this transition is consistent with previously reported shift responses to NETs [39].

Furthermore, the histopathological changes also proved the effects of DNase-1 on NPE caused by SAH. The markers representing the formation of NETs showed that the formation of NETs was significantly reduced after the treatment of DNase-I. Interestingly, we found that the neurological scores increased significantly in the DNase-I-treated mice, while there was no significant difference in the SAH scores between the DNase-1-treated mice and the mice without treatment after SAH. This finding may suggest that the remission of NPE through DNase-1 can increase the neurological scores and even improve the prognosis. Our study focused on the effect of the injection of DNase-1 on NPE 24 h after SAH in mice. However, the long-term effects of the DNase-1 treatment have not been proved.

Since the specific mechanism of NPE after SAH has not been clarified yet, the clinical administration of NPE remains a challenge. Previous studies found NETs involved in lung injuries. Therefore, we designed this study to identify the role of NETs in NPE after SAH and then found that NETs were involved in it. We inferred the potential role of NETs in NPE after SAH. Further research of the NET role in NPE after SAH benefits the guidance of clinical decisions. We also need to do more studies to clarify the curative effects of drugs targeting NETs in the long term. Moreover, the relationship between the key regulator PAD4 and the neutrophils has been studied in our previous research, but the mechanism needs further study [18]. In this study, we aimed to explore the role of DNase-1 in NPE after SAH via inhibiting NETs; thus, we did not discuss the other aspects.

In summary, our study suggested that NETs are involved in NPE after SAH, and DNase-1 treatment can attenuate a lung injury in NPE after SAH. DNase-1 may be a potential choice for relieving NPE after SAH with potential clinical value.

## Figures and Tables

**Figure 1 jcm-11-04349-f001:**
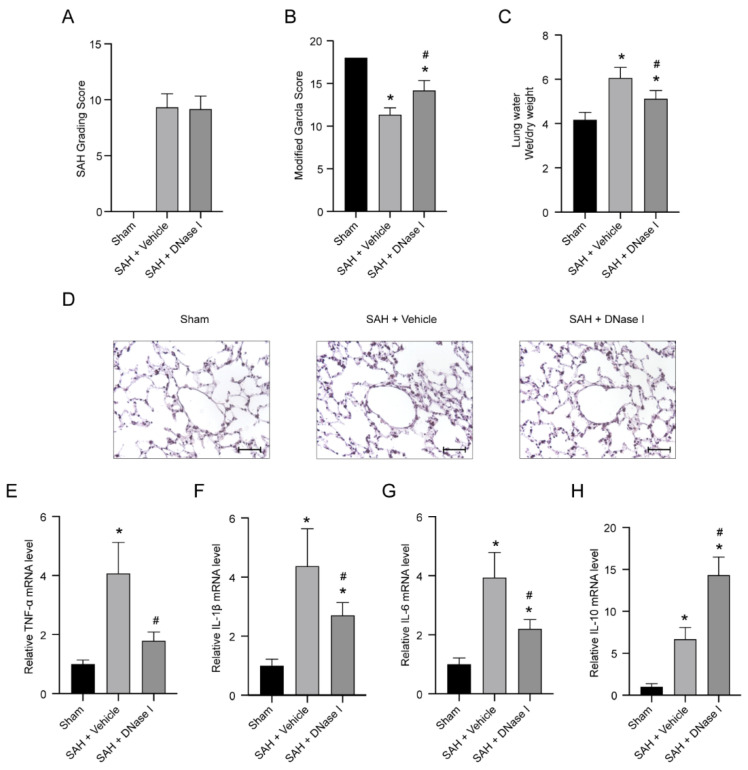
The protective effect of DNase-1 for neurogenic pulmonary edema (NPE) after subarachnoid hemorrhage (SAH). (**A**) The quantification of the SAH grade; *n* = 18/group. (**B**) The quantification of the modified Garcia scale; *n* = 18/group. (**C**) Quantification of the lung water; *n* = 6/group. (**D**) Representative lung histopathology at 24 h after SAH. Scale bar = 50 μm; *n* = 5/group. (**E**–**H**) Quantitative qPCR analysis of the relative expression of TNF-α, IL-1β, IL-6 and IL-10; *n* = 5/group. * *p* < 0.05 versus the sham and # *p* < 0.05 versus the SAH + vehicle group.

**Figure 2 jcm-11-04349-f002:**
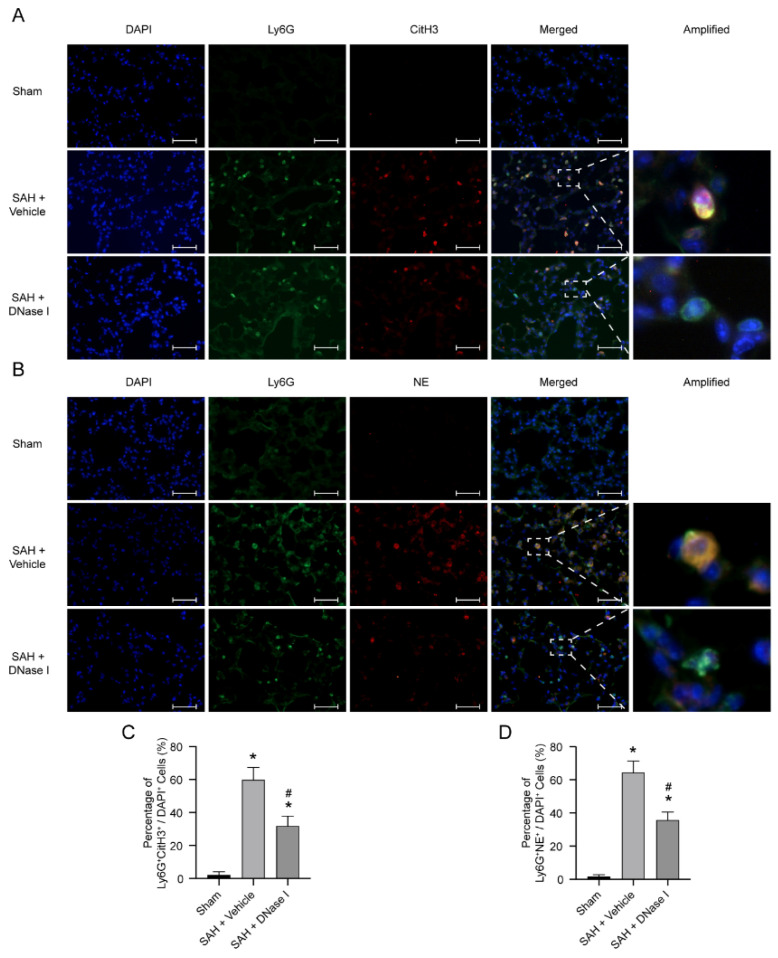
DNase-1 inhibits the formation of NETs in the NPE after SAH. (**A**) Representative photograph of the colocalization of CitH3 (red) with Ly6G (green) among the groups. The nuclei were stained with DAPI (blue). Scale bar = 50 μm. (**B**) Representative photograph of the colocalization of NE (red) with Ly6G (green) among the groups. The nuclei were stained with DAPI (blue). Scale bar = 50 μm. (**C**) Quantitative analysis of CitH3-positive neutrophils; *n* = 5/group. (**D**) Quantitative analysis of NE-positive neutrophils; *n* = 5/group. * *p* < 0.05 versus the sham and # *p* < 0.05 versus the SAH + vehicle group.

**Figure 3 jcm-11-04349-f003:**
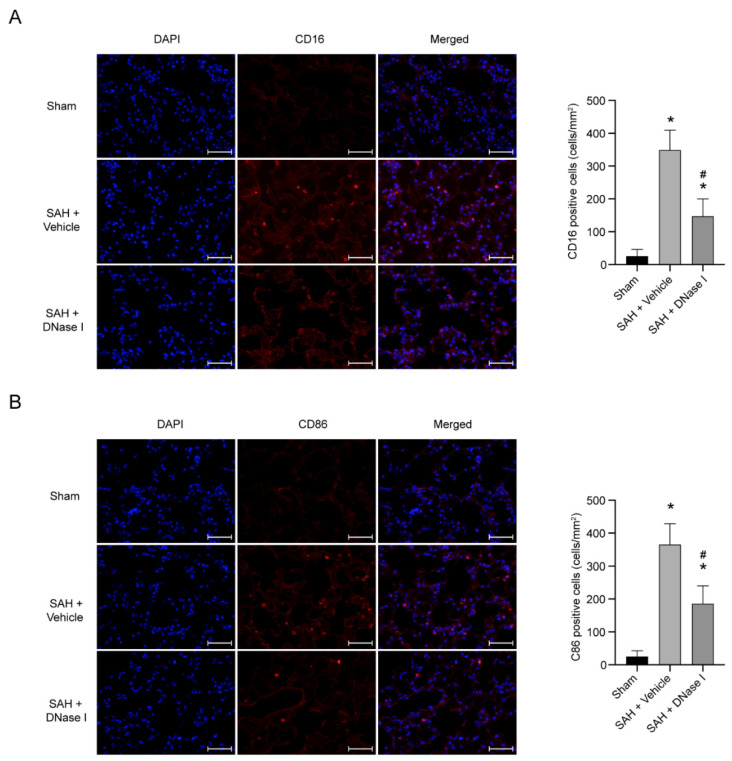
DNase-1 inhibits the proinflammatory transition in NPE after SAH. (**A**) Representative photograph and quantitative analysis of CD16-positive cells (red) among the groups. The nuclei were stained with DAPI (blue). Scale bar = 50 μm; *n* = 5/group. (**B**) Representative photograph and quantitative analysis of CD86-positive cells (red) among the groups. The nuclei were stained with DAPI (blue). Scale bar = 50 μm; *n* = 5/group. * *p* < 0.05 versus the sham and # *p* < 0.05 versus the SAH + vehicle group.

**Figure 4 jcm-11-04349-f004:**
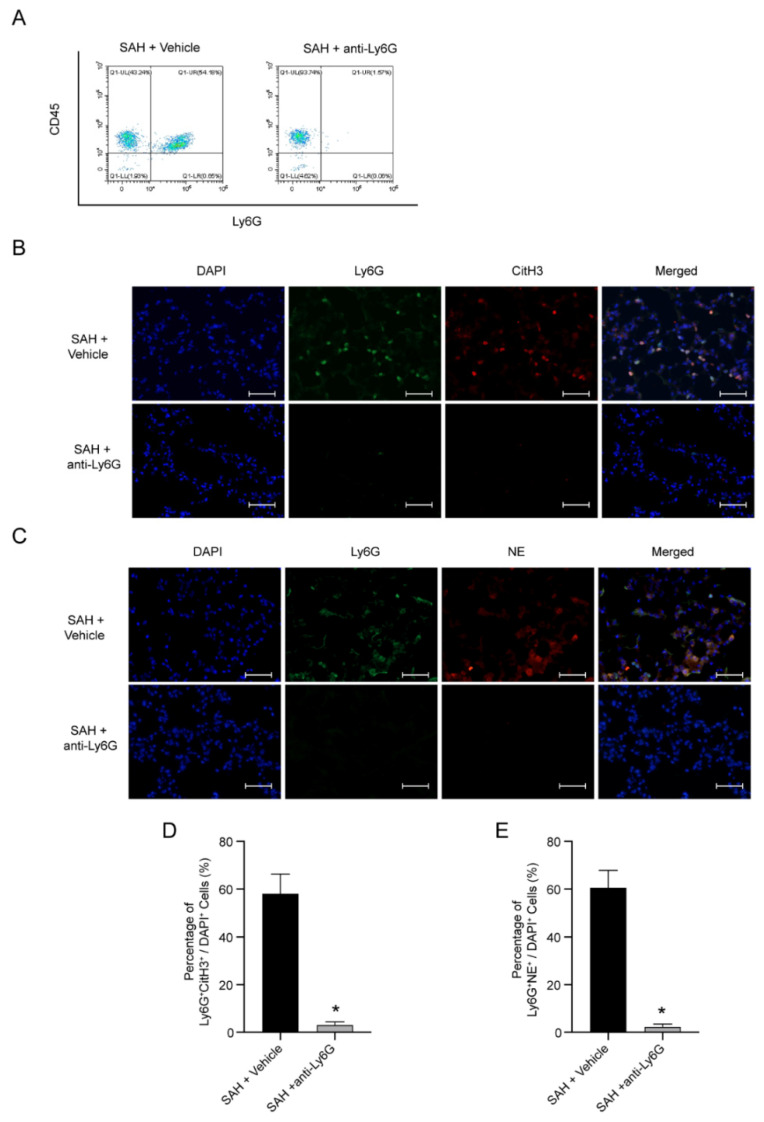
Ly6G depletion decreases the formation of NETs in NPE after SAH. (**A**) Representative flow cytometry analysis of the blood from SAH mice in different groups. (**B**) Representative photograph of the colocalization of CitH3 (red) with Ly6G (green) in different groups. The nuclei were stained with DAPI (blue). Scale bar = 50 μm. (**C**) Representative photograph of the colocalization of NE (red) with Ly6G (green) in different groups. The nuclei were stained with DAPI (blue). Scale bar = 50 μm. (**D**) Quantitative analysis of CitH3-positive neutrophils; *n* = 5/group. (**E**) Quantitative analysis of NE-positive neutrophils; *n* = 5/group. * *p* < 0.05 versus the SAH + vehicle group.

## Data Availability

Not applicable.

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
