# Peer review of "DNase-1 Treatment Exerts Protective Effects in Neurogenic Pulmonary Edema via Regulating the Neutrophil Extracellular Traps after Subarachnoid Hemorrhage in Mice"

_jcm, 2022, doi:10.3390/jcm11154349_

Round 1

Reviewer 1 Report

 In this study, Wu et al. showed that NETosis might participate in SAH-induced NPE injury and DNase-1 treatment has the potential to relieve the NPE after SAH effectively and to be a clinical drug after SAH. The paper is well-written, the results are potentially interesting, and the method looks appropriate.

Several essential concerns should be addressed to improve the overall impact of this study.

Major concern:

  1. In this article, only 24 hours post-SAH data is shown. It's important to demonstrate the expression profile of NETs in SAH-induced NPE.  
  2. Only CitH3 was used as a NETs marker in this study. The relationship between PADs, neutrophils, etc., is not discussed.
  3. The design of this study is very similar to that of another published study (Translational Stroke Research volume 13, pages112–131, 2022). It needs to be further explained whether different tissues (brain vs. lung) were tested with the same batch of animals.
  4. The expression of cytokines (PCR) and IHC is insufficient to clarify the NETosis in SAH-induced NPE; protein expression is relatively important for quantification. 
  5. IHC and PCR alone are not enough to clarify the role of NETosis in SAH-induced NPE, and it is necessary to analyze protein changes by immunoblotting.

Minor concern:

  1. It is confused about the 2nd DNaseI administration route and the timepoint.
  2. Fig. 1C Dry/Wet Method of Lung Edema without Unit?
  3. Inappropriate or misspelling in text:

         Line 90: .[REF]

         Consistency: 

              24 h vs. 24 hr vs. 24-hr.

              P-value: P<0.05 vs. p<0.05

          Line 146: Gacia to Garcia

          Line 175 Font size of the Subtitle.

Author Response

  • Thank you for your review. According to the previous research, 24 hours after SAH is the common point of time (Brain : a journal of neurology volume 124(Pt 2), pages 249-78, 2001) (Journal of pineal research volume 59(4), pages 469-77, 2015). Meanwhile, this point of time was verified in our pre-experiments. Our research tends to provide a certain basis for the clinical transformation of NPE after DNase I treatment for SAH in the future, thus we chose the point of time where NPE is relatively obvious.
  • Another published study (Translational Stroke Researchvolume 13, pages112–131, 2022) you mentioned before has preliminarily discuss the relationship between PAD and neutrophils. This study focused more on the role of DNase I inhibiting NETs in NPE resulted from SAH.
  • The study was not designed to use the same animals as another published study (Translational Stroke Researchvolume 13, pages112–131, 2022). The two projects were carried out by different research groups. Based on the expertise of researcher HHZ in this field, we invited him to give guidance to this project, but the main work was still completed by XYW and YHG.
  • We have supplemented the protein expression of NETs marker in the supplementary of the revised draft. Thank you for your comments.
  • We have supplemented the protein expression of NETs marker in the supplementary of the revised draft. Thank you for your comments.

Minor

  • The drug-delivery way refers to the methods in the previous reference(Stroke volume 50(11), pages 3228-3237, 2019)(P2, 73-74).
  • 1C Dry/Wet Method of Lung Edema is ratio, so we represented without unit.
  • We are grateful to the viewers’ comments and we have changed accordingly in the revised manuscript.

Reviewer 2 Report

This is an interesting article addressing an important issue in subarachnoid hemorrhage reserach. I do have only two important points that the authors are advised to revise. First, I think it is important to discuss the animal model used and why the authors used the reported. Standard parameters in SAH animal models have been identified and deviation to this should be discussed in the manuscript. Second, there is a need to report according to the ARRIVE guidelines (for example you have to report the mortality and morbidity rate - and again, your rate should be compared to those reported in the literature - please refer to "standard animal models of SAH").

Author Response

  1. The reasons for choosing this model have been discussed in the revision (P10, 226-236). Thank you for your suggestions.
  2. We are grateful to the viewers’ comments. We have supplemented relevant data in the revision (P4, 141-146) and compared with the previous references in discussion (P10, 234-235).

Round 2

Reviewer 1 Report

The authors reply to all my concerns.